# Evaluation of a Sequential Antibiotic Treatment Regimen of Ampicillin, Ciprofloxacin and Fosfomycin against *Escherichia coli* CFT073 in the Hollow Fiber Infection Model Compared with Simultaneous Combination Treatment

**DOI:** 10.3390/antibiotics11121705

**Published:** 2022-11-26

**Authors:** Ashok Krishna, Tesfalem Zere, Sabyasachy Mistry, Omnia Ismaiel, Heather Stone, Leonard V. Sacks, James L. Weaver

**Affiliations:** 1Division of Applied Regulatory Science, Office of Clinical Pharmacology, Center for Drug Evaluation and Research, US Food and Drug Administration, Silver Spring, MD 20993, USA; 2Office of Product Evaluation and Quality, Center for Devices and Radiological Health, US Food and Drug Administration, Silver Spring, MD 20993, USA; 3Office of Medical Policy, Center for Drug Evaluation and Research, US Food and Drug Administration, Silver Spring, MD 20993, USA

**Keywords:** antibiotics, antibiotic resistance, hollow fiber infection model, sequential antibiotic regimen, *E. coli* CFT073

## Abstract

Objective: Employ the hollow fiber infection model (HFIM) to study sequential antibiotic administration (ampicillin, ciprofloxacin and fosfomycin) using human pharmacokinetic profiles to measure changes in the rate of antibiotic resistance development and compare this to simultaneous combination therapy with the same antibiotic combinations. Methods: Escherichia coli CFT073, a clinical uropathogenic strain, was exposed individually to clinically relevant pharmacokinetic concentrations of ampicillin on day 1, ciprofloxacin on day 2 and fosfomycin on day 3. This sequence was continued for 10 days in the HFIM. Bacterial samples were collected at different time points to enumerate total and resistant bacterial populations. The results were compared with the simultaneous combination therapy previously studied. Results: Sequential antibiotic treatment (ampicillin-ciprofloxacin-fosfomycin sequence) resulted in the early emergence of single and multi-antibiotic-resistant subpopulations, while the simultaneous treatment regimen significantly delayed or prevented the emergence of resistant subpopulations. Conclusion: Sequential administration of these antibiotic monotherapies did not significantly delay the emergence of resistant subpopulations compared to simultaneous treatment with combinations of the same antibiotics. Further studies are warranted to evaluate different sequences of the same antibiotics in delaying emergent resistance.

## 1. Introduction

Antibiotics are used to treat bacterial infections, and their efficacy is threatened by the rapid emergence of resistant bacteria worldwide [1]. Bacterial infections caused by Gram-negative bacteria like Enterobacteriaceae, Pseudomonas and Acinetobacter species are becoming resistant to most antibiotic drug options available [2,3]. The antibiotic resistance problem has been associated with the overuse or misuse of antibiotics, and the small number of new antibiotics being developed provides limited options for treating resistant bacteria.

With the growing need to combat the antibiotic resistance crisis, it has become necessary to reconsider the monotherapy standard and explore combination therapies of existing antibiotics or antibiotics with non-antibiotic enhancing compounds to diminish the emergence of spontaneous resistance [4]. To test these combination therapies, nonclinical infection models are commonly used to derive pharmacokinetic-pharmacodynamic (PK/PD) relationships of antibacterial combinations and provide critical information for designing optimal human dosage regimens [5,6]. Both in vitro and in vivo models are helpful for large-scale screening of synergy of antibiotic combinations, and the results obtained can be further validated in prospective clinical studies [7]. Among the in vitro models, the hollow fiber infection model (HFIM) has become an important tool in exploring the efficacy of antibiotics to prevent emergent antibiotic resistance and to combat serious infections caused by multidrug-resistant bacteria [8,9,10].

With bacterial mutations, new mechanisms evolve to survive specific antibiotic treatment and thereby diminish the antibiotic efficacy in treating bacterial infection. Different treatment strategies using combination and sequential therapy have been investigated to reduce the emergence of antibiotic resistance. Antibiotic combinations are also increasingly employed to enhance the antibacterial effects of available drugs against multidrug strains. Compared to monotherapy, the advantage of using combinations includes a broader spectrum, synergistic effect, and reduced risk of emerging resistance during therapy [11,12,13]. Several in vitro studies using double or triple antibiotic combinations against multidrug-resistant *Pseudomonas* spp., *Acinetobacter* spp., and carbapenemase-producing *Enterobacteriaceae* have demonstrated synergistic effects [14,15,16,17,18]. However, some antibiotic combinations can accelerate resistance emergence through gene amplification of efflux pumps [19]. This undesired effect is avoided by the sequential application of antibiotics, where the negative hysteresis and collateral sensitivity increases the killing efficiency of other antibiotics in sequence [20]. However, the efficacy of sequential treatment, with either a single switch between antibiotics or multiple switches at short intervals, depends both on the antibiotic classes included and the particular sequence of antibiotics in treatment [21,22].

In our previous study, we utilized the HFIM to study the effect of single, double, or triple simultaneous combination treatments of ampicillin, ciprofloxacin and fosfomycin antibiotics in reducing the total viable population of bacteria, as well as the effect of antibiotic combinations on the emergence of resistant subpopulations using the Escherichia coli CFT073 strain [23]. The study results showed that simultaneous double or triple antibiotic combinations significantly delayed the emergence of resistant bacteria compared to single antibiotic treatments alone. In the present study, ampicillin, ciprofloxacin and fosfomycin are administered sequentially. This sequential therapy is studied for 10 days for its effect on total and resistant subpopulations in the HFIM. The primary aim of this study is to compare the efficacy of sequential treatments with the combination treatment studied previously in delaying emergent resistance. The schematics representing the study objective are given below (Figure 1).

## 2. Results

### 2.1. Antibiotic Treatment Effect on Total E. coli Population

To characterize the pharmacodynamic activity of the sequential treatment strategy against the *E. coli* CFT073 strain, we simulated human pharmacokinetic exposure to ampicillin, ciprofloxacin and fosfomycin (ACF) in a ten-day HFIM. On day 1 of ampicillin administration, a reduction of about 3 to 4 log_10_ cfu/mL in the total bacterial population was observed after 4 h. At 24 h post-ampicillin treatment, the total bacterial population increased from the initial inoculum of approximately 10^8^ cfu/mL to 10^10^ cfu/mL. Further treatment with ciprofloxacin on day 2, fosfomycin on day 3 and subsequent sequential therapy until 10 days did not affect the total bacterial population (Figure 2A–C).

### 2.2. Single Antibiotic-resistant E. coli Subpopulations

Over 10 days, we studied the effect of ampicillin, ciprofloxacin and fosfomycin given on sequential days on the emergence of bacterial subpopulations of *E. coli* resistant to 3×, 10× and 30× MIC of each antibiotic, and to their double and triple antibiotic combinations. The emergence of the subpopulations resistant to 3×, 10× and 30× MIC of each individual antibiotic are illustrated in Figure 2. Bacterial subpopulations resistant to 3×, 10× and 30× MIC of each of the three antibiotics appeared within 24 h after each antibiotic treatment was started, except for subpopulations resistant to 30× MIC ampicillin, which appeared at 48 h. Subpopulations resistant to 3× MIC of ampicillin and ciprofloxacin constituted most of the total population (9 to 10 Log_10_ cfu/mL). Subpopulations resistant to 3× MIC of fosfomycin were present at about 6 to 7 Log_10_ cfu/mL. The 10× MIC ciprofloxacin-resistant subpopulations (9 to 10 Log_10_ cfu/mL) constituted most of the total population, followed by ampicillin (5 to 8 Log_10_ cfu/mL) and fosfomycin (2 to 3 Log_10_ cfu/mL) subpopulations. The 30× MIC resistant subpopulations were observed at the following concentrations: ciprofloxacin (7 to 9 Log_10_ cfu/mL), ampicillin (2 to 4 Log_10_ cfu/mL) and fosfomycin (2 to 3 Log_10_ cfu/mL) (Figure 2A–C).

### 2.3. Double Antibiotic-Resistant E. coli Subpopulations

The emergence of subpopulations simultaneously resistant to 3×, 10× and 30× MIC for combinations of ampicillin–ciprofloxacin, ampicillin–fosfomycin and ciprofloxacin–fosfomycin are shown in Figure 3A–C. Following the exposure of the bacterial population to ampicillin on day 1 and ciprofloxacin on day 2, subpopulations resistant to both ampicillin and ciprofloxacin appeared 24 h after the introduction of ciprofloxacin and subpopulations simultaneously resistant to ampicillin and fosfomycin and ciprofloxacin and fosfomycin appeared 24 h after the introduction of fosfomycin. A subpopulation resistant to both 10× MIC of ampicillin and ciprofloxacin constituted most of the double antibiotic-resistant subpopulations, followed by a subpopulation resistant to both ampicillin and fosfomycin. Subpopulations resistant to both ciprofloxacin and fosfomycin appeared only at the 3× MIC level, and no subpopulations resistant to both ciprofloxacin and fosfomycin appeared at the 10× MIC level. No subpopulations simultaneously resistant to 30× MIC for any of the double antibiotic combinations were observed (Figure 3A–C).

### 2.4. Triple Antibiotic-resistant E. coli Subpopulations

Twenty-four hours after the first cycle of sequential antibiotic therapy, colonies were found that were simultaneously resistant to 3× MIC of ampicillin, ciprofloxacin, and fosfomycin (Figure 4A). No colonies were detected with simultaneous resistance to the 10× and 30× MIC levels of the three antibiotics (Figure 4A).

### 2.5. Effect of Single Antibiotics on Total and 3× MIC Resistant E. coli Subpopulations

Treatment with individual antibiotics (ampicillin, ciprofloxacin and fosfomycin) resulted in a reduction of the total bacterial population for the first 6 h but had no effect on days 2 and 3 (Figure 5). Colonies resistant to 3× MIC of ampicillin and 3× MIC of fosfomycin only appeared at 6- and 24-h following treatment (Figure 5A, C). In contrast, colonies resistant to 3× MIC of ciprofloxacin appeared at 4 h post-treatment on day 1 (Figure 5B). By days 2 and 3, none of the antibiotics showed any effect on subpopulations resistant to 3× MIC of each antibiotic (Figure 5A–C).

## 3. Discussion

Antibiotic resistance in Gram-negative bacterial pathogens represent a serious threat to public health and results in significant morbidity and mortality worldwide. The World Health Organization (WHO) has published a list of antibiotic-resistant pathogens, and among them, the most common are Gram-negative bacteria [3,24]. To address the widespread emergence of antibiotic-resistant strains, global research efforts are underway to develop new antibiotics, as well as to modify or repurpose existing antibiotics [25]. Our research efforts focused on assessing emergent resistance following treatment with three antibiotics commonly used as monotherapy in the management of urinary tract infections (ampicillin, ciprofloxacin and fosfomycin). We previously studied the effects of simultaneous combination therapy on emergent resistance in *E. coli* CFT073 using a hollow fiber infection model (HFIM). In this paper, we describe the effects of sequential monotherapy using the same model and compare the results to simultaneous administration.

Our earlier research showed that simultaneous antibiotic combinations resulted in the sustained killing of the total bacterial population. Only a few viable colonies were observed between 24–96 h, which gradually increased to 10^10^ cfu/mL on day 10. Also, simultaneous combination antibiotic therapy significantly delayed the emergence of any resistant *E. coli* CFT073 subpopulations compared to the monotherapies. The resistant subpopulations that eventually emerged after combination administration generally showed low-grade resistance (3× MIC) compared to monotherapy, and there was no appreciable outgrowth of multidrug-resistant colonies resistant to all three antibiotics. The emergence of single drug-resistant colonies for all three antibiotics at 3× and 10× MIC were delayed until 96–120 h and 144–168 h, respectively [23]. In contrast, sequential monotherapy resulted in the rapid appearance of resistant colonies following the introduction of each antibiotic. The total bacterial load recovered soon after introducing the first antimicrobial in the sequence, reaching baseline concentrations of 10^10^ cfu/mL after 24 h. Surprisingly, introducing a second and third antimicrobial in sequence did not appreciably reduce the total bacterial load any further, possibly as the result of the initial induction of efflux pumps or other non-specific resistance mechanisms.

During sequential therapy, subpopulations with high-grade monoresistance (30× MIC) to each added antibiotic emerged rapidly. In contrast, following simultaneous combination therapy, high-grade monoresistance was observed for fosfomycin at 168 h post-treatment, but not for ampicillin or ciprofloxacin. When we studied the emergence of multidrug resistance, we found that the sequential antibiotic regimen resulted in the cumulative appearance of single, double, and triple antibiotic-resistant subpopulations following the introduction of each new antibiotic, frequently at high grade (10× MIC). Resistance to more than one drug was rare following simultaneous combination therapy and was low grade (3× MIC) when it did appear. Resistant populations to more than one antibiotic appeared for each of the two antibiotic combinations after 192 to 216 h and for the three antibiotic combinations after 192 h.

Our findings on sequential monotherapy showed an accumulation of cumulative and often high-grade multidrug resistance. An in vitro study with quinolones, β-lactams, and aminoglycosides with isolates of commonly encountered nosocomial pathogens showed the development of multiple drug resistance to related and unrelated classes of antibiotics. This unusual pattern of multiple drug resistance was shown to be associated with changes in the outer membrane proteins of bacterial pathogens [26]. Sequential monotherapy frequently occurs in clinical practice and is usually associated with changing antibiotic resistance profiles. In a clinical study, two different antibiotic classes (quinolone and β-lactam) were cycled for four 4-month periods in a surgical intensive care unit. The potential for selection of antibiotic-resistant Gram-negative bacteria, during periods of homogenous exposure, increased from cefpirome to piperacillin/tazobactam to levofloxacin. Study results showed that cycling of homogenous antibiotic exposure is unlikely to control the emergence of Gram-negative antimicrobial resistance. Another study with stochastic modeling of accumulated clinical trial data over a period showed that neither antibiotic cycling nor antibiotic mixing is better than the other at mitigating for selection of antibiotic resistance in the clinic. [27,28].

Simultaneous treatment with three antibiotics aimed at different bacterial targets constitutes a strong selection pressure and appears to disallow the emergence of significantly resistant mutants. Combination treatments with three antibiotics are gaining acceptance to treat highly resistant pathogens based on their ability to target multiple targets in the bacteria. Several studies, including our earlier study, showed that the administration of single antibiotics resulted in the appearance of resistant bacteria, while combinations reduced their emergence [4,23,25]. The increased killing of bacteria with our use of combination therapy may result from the action of ampicillin binding to the penicillin-binding protein to inhibit bacterial cell wall synthesis and fosfomycin’s interference with bacterial cell wall biosynthesis. Furthermore, disrupting the integrity of the outer membrane, and consequently enabling ciprofloxacin entry, enhances the target site action on topoisomerase II (DNA gyrase) and topoisomerase IV.

The differences observed between the sequential and simultaneous treatment regimens may depend on several factors, such as different mechanisms of action of each drug administered, the sequence of drugs administered, the time interval between each antibiotic administration and the type of resistance that occurs. In the sequential study, ampicillin was administered while the bacteria were in the logarithmic growth phase on day 1. Further treatments with ciprofloxacin on day 2 and fosfomycin on day 3 were administered to the static total bacterial population, along with resistant subpopulations. In the case of simultaneous treatment, all three antibiotics were administered to an actively growing susceptible bacterial population. The antibiotic action on the different growth phases of the bacterial population, along with the generation of resistant subpopulations, might have resulted in the differences observed in the accumulation of resistant mutants between these two treatment regimens. Further hollow fiber experiments with individual antibiotics given for 3 days showed that the effects during the initial 6 h after administration were similar to those observed with sequential therapy. Each individual antibiotic reduced the total bacterial population on day 1, but continued administration of the same antibiotics on days 2 and 3 had no effect on the total bacterial population. The resistance pattern observed with sequential therapy (ampicillin-ciprofloxacin-fosfomycin) might have been different if the same antibiotics had been used in different sequences. This has been observed in several studies conducted with different regimens of sequential treatments involving either a single switch between antibiotics or multiple switches at short intervals depending on both the included antibiotic classes and the particular sequence of antibiotics in treatment. The main findings of these studies were that the efficacy of the sequential treatments depended on both the antibiotic classes and the treatment sequence [29,30,31].

The emergence of new genetic resistance is a major public health concern and has resulted in many fatal outbreaks [32,33,34,35]. Once established, resistant organisms become a permanent feature of the bacterial ecosystem, resulting in increased morbidity, mortality and a loss of effective antibacterial agents. Clinical stewardship has been widely implemented to minimize this, but other strategies are clearly needed. Our study examined phenotypic resistance, which may be the product of genetic mutations, transmitted resistance elements, multidrug efflux pumps, such as AcrAB-TolC and Mex pumps of the resistance-nodulation-division (RND) superfamily, biofilms, bacterial stress responses virulence factors expressed by *E. coli* cells [36,37], as well as other mechanisms, either alone or in combination. Whole genome sequencing studies are under way to investigate the genotypic mechanisms of resistance underlying the phenotypic resistance that we have documented.

## 4. Materials and Methods

### 4.1. Microorganisms, Antibiotics, Media, and Agar

The clinical *Escherichia coli strain* CFT073-WAM4505 (ATCC BAA-2503) (strain isolated from the blood of a patient with pyelonephritis) purchased from American Type Culture Collection, ATCC, (Manassas, VA, USA) was used for experimental infection in the hollow fiber infection model for antibiotic resistance development studies. This strain was grown in Luria-Bertani (LB) broth to a logarithmic growth phase; 15% glycerol stocks were made and stored at −70 °C. Prior to each experiment, this strain was streaked for isolation onto LB agar plates incubated overnight at 37 °C. A single colony was isolated, inoculated into 10 mL of LB broth and incubated at 37 °C overnight. The overnight culture was then subcultured (1:100) the next day into fresh LB broth, incubated at 37 °C and was grown until it reached the mid-exponential growth phase. Bacterial growth was determined by measuring the optical density of the culture at 600 nm (OD_600_). The mid-exponential phase culture (OD_600_ ~ 0.5; ~10^8^ CFU/mL) was used as a starting inoculum for the hollow-fiber experiment.

Ampicillin sodium salt, ciprofloxacin hydrochloride and fosfomycin disodium salt were obtained from Sigma-Aldrich, St. Louis, MO, USA. Stock solutions of the antibiotics were prepared in sterile water and filter sterilized. LB broth and agar were obtained from bioWORLD (Dublin, OH, USA). Sterile LB broths and agar were prepared by autoclaving, and filter-sterilized antibiotic solutions were used for the HF experiments.

### 4.2. In Vitro Hollow Fiber Infection Model

In vitro HFIM experiments were conducted for 10 days as previously described by Garimella, Zere et al., 2020 [23] with slight modifications in the treatment regimens. In brief, polysulfone cartridges with a 20 kD pore size (C2011, FiberCell Systems Inc., Frederick, MD, USA) were used in the HFIM system and were maintained at 37 °C in a humidified incubator. An initial inoculum, 20 mL of approximately 10^8^ CFU/mL mid-log phase *Escherichia coli* culture, prepared as described in Section 2.1, was aseptically inoculated into the extra capillary space of the polysulfone cartridges of the HFIM and was exposed to the sequential treatment of the three antibiotics as described below. With the use of infusion pumps, ampicillin (amp) (40 µg/mL; thrice daily with 8 hr intervals), ciprofloxacin (cip) (0.6 µg/mL; once daily) and fosfomycin (fos) (470 µg/mL; once daily) were pumped directly into the central reservoir of the hollow fiber assembly to reach clinically achievable concentrations (C_max_ of 10 µg/mL, 0.16 µg/mL and 100 µg/mL for amp, cip and fos respectively) in the hollow fiber cartridge by simulating human free drug pharmacokinetic profiles. The antibiotics were administered individually each day in the following order for 10 days: amp (day 1), cip (day 2), fos (day 3), amp (day 4), cip (day 5), fos (day 6), amp (day 7), cip (day 8), fos (day 9) and amp (day 10). The intended PK profiles achieved in the HFIM were confirmed by sampling at different time points (0, 5, 10, 20, 30, 45 and 60 min; 1, 2, 3, 4 and 24 h) and quantified as previously described [23]. In a separate study, three-day HFIM experiments were conducted to study the effect of single antibiotic treatment on the total and 3× resistant subpopulations during the initial 6 h of post-antibiotic treatment each day for 3 days.

### 4.3. Assessment of Microbiological Response

Bacterial samples (1–2 mL) from the sequential antibiotic therapy HF experiments were collected aseptically from the cartridges for the determination of total viable and antibiotic-resistant subpopulations at baseline, and 1, 2, 3 and 4 h on the first day of antibiotic treatment, then once daily at 24 h intervals for ten days. For the three-day single antibiotic HF experiments, bacterial samples (0.5 mL) were collected 1, 2, 3, 4, 5 and 6 h post-antibiotic treatments each day for 3 days. Serially diluted samples in LB broth were quantitatively cultured onto drug-free LB agar plates to enumerate the total viable bacterial population. A portion of the bacterial suspension was also quantitatively cultured onto LB agar, supplemented with either ampicillin or ciprofloxacin or fosfomycin or their double or triple combinations at 3×, 10× and 30× the baseline minimum inhibitory concentrations (MIC). This was performed to assess the impact of the treatment regimen on the development of resistant subpopulations. LB agar plates were incubated at 37 °C, and viable colonies were counted after 24 h of incubation. For some antibiotic combinations at 10× and 30× MIC levels, the plates were incubated for 48 to 72 h to enumerate the less susceptible resistant subpopulations.

### 4.4. Statistical Analysis

The total population and resistant subpopulation bacterial counts geometric mean and geometric standard deviation were determined from three independent sequential therapy HFIM experiments by using GraphPad Prism 8 software.

## 5. Conclusions

This study compared the efficacy of a sequential antibiotic treatment regimen (ampicillin-ciprofloxacin-fosfomycin) to simultaneous combination therapy in reducing the emergence of resistant subpopulations using a hollow fiber infection model. Sequential antibiotic treatment resulted in the appearance of single and multi-antibiotic-resistant subpopulations, but simultaneous treatment significantly delayed or prevented the emergence of resistant subpopulations. Further studies that evaluate different sequences of the same antibiotics in delaying emergent resistance are warranted.

## Figures and Tables

**Figure 1 antibiotics-11-01705-f001:**
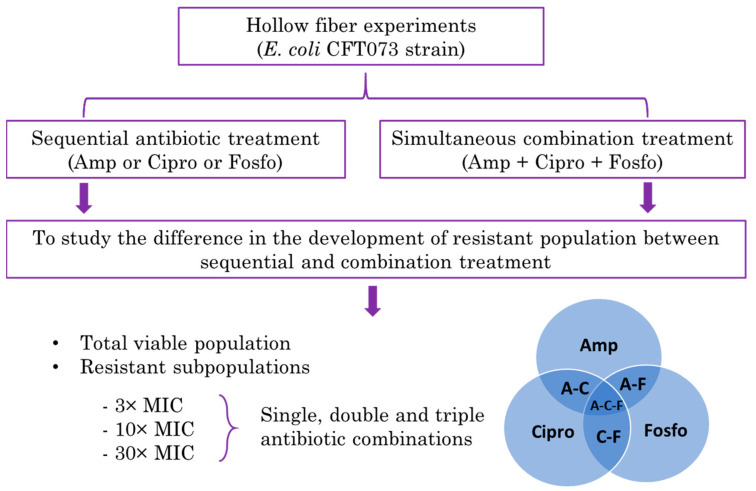
Schematics representing the hollow fiber infection model (HFIM) study comparing the sequential antibiotic treatment with the combination antibiotic treatment in delaying the resistance emergence.

**Figure 2 antibiotics-11-01705-f002:**
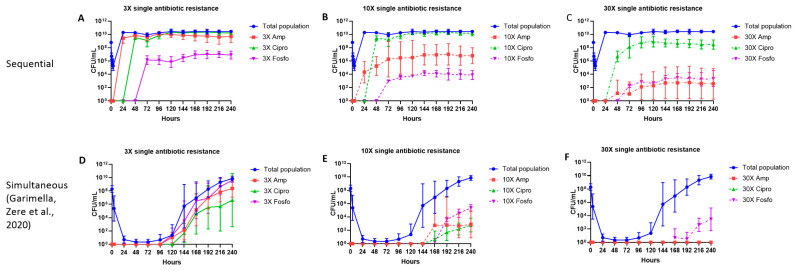
The effect of ampicillin, ciprofloxacin and fosfomycin sequential (**A**–**C**) and simultaneous combination (**D**–**F**) [23] therapy regimens on the total population and single antibiotic-resistant subpopulations in the hollow fiber infection model over 10 days. The 3×, 10× and 30× single antibiotic-resistant subpopulations were grown on agar containing the respective antibiotic minimum inhibitory concentration (MIC) levels. Data are presented as the geometric mean of three (Sequential regimen), and four (Simultaneous combination regimen) independent hollow fiber runs, and the error bars represent the geometric standard deviation.

**Figure 3 antibiotics-11-01705-f003:**
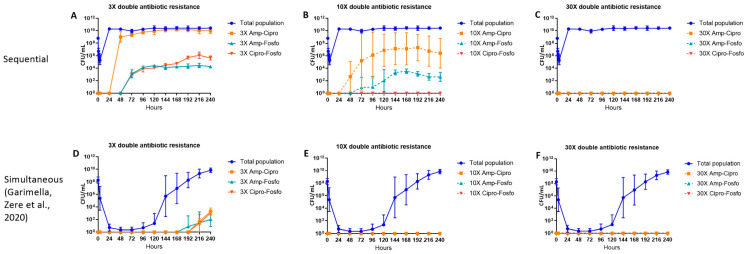
The effect of ampicillin, ciprofloxacin and fosfomycin sequential (**A**–**C**) and simultaneous combination (**D**–**F**) [23] therapy regimens on the density of the total and double antibiotic-resistant subpopulations in the hollow fiber infection model over 10 days. The 3×, 10× and 30× double antibiotic-resistant subpopulations (Amp-Cipro, Amp-Fosfo and Cipro-Fosfo) were grown on agar containing the respective combinations of antibiotic minimum inhibitory concentration (MIC) levels. Data are presented as the geometric mean of three (sequential regimen) and four (simultaneous combination regimen) independent hollow fiber runs, and the error bars represent the geometric standard deviation.

**Figure 4 antibiotics-11-01705-f004:**
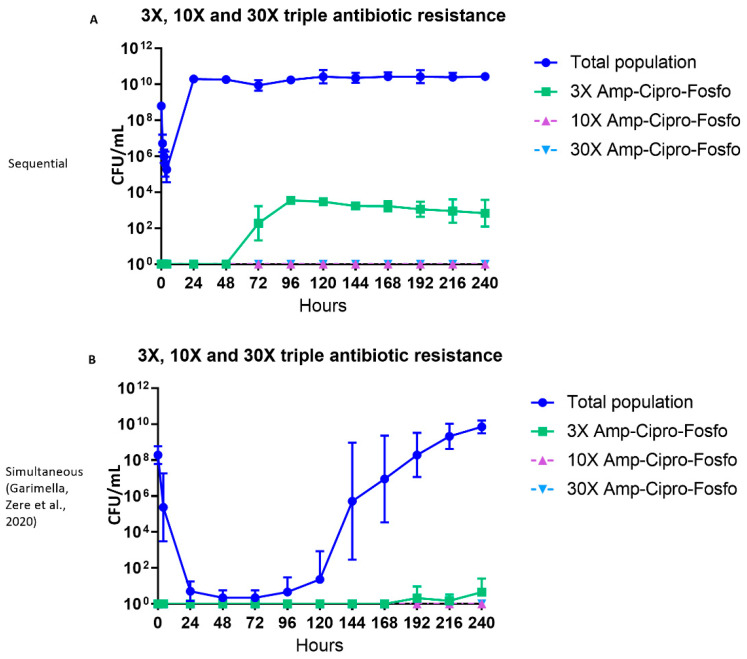
The effect of ampicillin, ciprofloxacin and fosfomycin sequential (**A**) and simultaneous combination (**B**) [23] therapy regimens on the density of the total and triple antibiotic-resistant subpopulations in the hollow fiber infection model over 10 days. The 3×, 10× and 30× triple antibiotic-resistant subpopulations (Amp-Cipro-Fosfo) were grown on agar containing the respective combinations of antibiotic minimum inhibitory concentration (MIC) levels. Data are presented as the geometric mean of three (sequential regimen) and four (simultaneous combination regimen) independent hollow fiber runs, and the error bars represent the geometric standard deviation.

**Figure 5 antibiotics-11-01705-f005:**
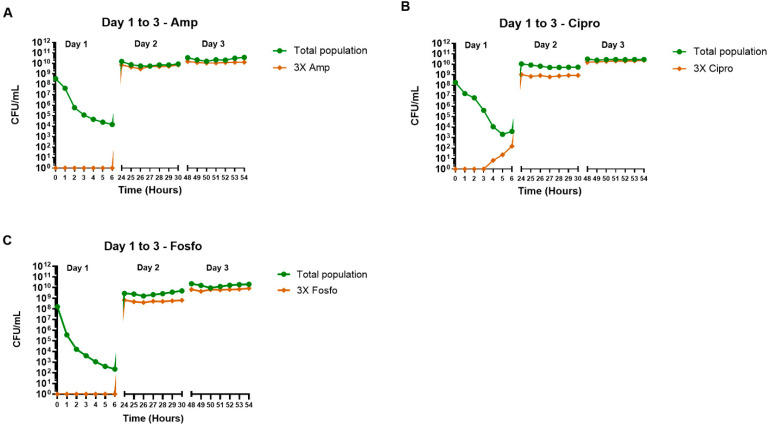
The effect of ampicillin (**A**), ciprofloxacin (**B**) and fosfomycin (**C**) single antibiotic therapy regimens on the density of the total and 3× antibiotic-resistant subpopulations in the hollow fiber infection model. The samples were collected up to 6 h post-antibiotic treatment over 3 days. The 3× antibiotic-resistant subpopulations were grown on agar containing the respective antibiotic minimum inhibitory concentration (MIC) levels. Data shown are from single independent hollow fiber runs.

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
