# Peer review of "Evaluation of a Sequential Antibiotic Treatment Regimen of Ampicillin, Ciprofloxacin and Fosfomycin against Escherichia coli CFT073 in the Hollow Fiber Infection Model Compared with Simultaneous Combination Treatment"

_antibiotics, 2022, doi:10.3390/antibiotics11121705_

Round 1

Reviewer 1 Report

The manuscript by Krishna, et al., “Evaluation of a sequential antibiotic treatment regimen of ampicillin, ciprofloxacin and fosfomycin against Escherichia coli CFT073 in the hollow fiber infection model compared with simultaneous combination treatment” explored an interesting and important question. The manuscript studied the sequential antibiotic administration by using human pharmacokinetic profiles to measure changes of antibiotic resistance. The results showed that sequential antibiotic treatment regimen had minimal effect on delaying resistant subpopulations. The research method is not very rigorous and the discussion is not very sufficient. I think some suggestion may be beneficial for improving the manuscript.

1.     As an important method used in the present study, the ‘sequential antibiotic therapy’ Method and its effect should be addressed in the introduction section.

2.     Although a serial of work was published before by the authors, the present manuscript should discuss the reason why the sequential antibiotic therapy was used and why it is appropriate for the present study; the present manuscript should discuss is there any other factor, interaction of combination drug or interaction of different drug metabolism products in simultaneous combination treatment, that effect the results?

3.     In all figures, I think the Y-axis should be labeled by CFU/mL. if the log10 were added as the present manuscript, the number of Y-axis should be 0, 1, 2, …, 12. It is appropriate to have two subgraphs per row.

4.     A, B and C, D, E, and F were all missed in the subplot of Figures. Further, why the geometric standard deviation of total population in some subplots were much smaller than their sub-components? What do these results mean? 

5.     Format should be checked throughout the manuscript. Some places have extra Spaces or some sentences lack periods. Line16,line31 and line33 in Page8,line82 in page9,author names were not correctly listed (reference list 14, 18, 19).

Author Response

The authors would like to thank the reviewers for their valuable comments to improve the quality of the manuscript. The reviewer’s concerns were considered and incorporated changes accordingly in the revised manuscript. The changes in the revised manuscript were highlighted by track changes.

1. As an important method used in the present study, the ‘sequential antibiotic therapy’ method and its effect should be addressed in the introduction section.

Many thanks for this comment. We now added more information on sequential antibiotic therapy and its effect on the development of resistant population in the introduction section (See page 2).

2. Although a serial of work was published before by the authors, the present manuscript should discuss the reason why the sequential antibiotic therapy was used and why it is appropriate for the present study; the present manuscript should discuss is there any other factor, interaction of combination drug or interaction of different drug metabolism products in simultaneous combination treatment, that effect the results?

Many thanks for this interesting point. We used E. coli in the hollow fiber infection model (HFIM) as an established pathogen model system for studying the rate of resistance development against antibiotics. We studied the commonly used antibiotics ampicillin, ciprofloxacin and fosfomycin for treating the urinary tract infections for resistance development. The focus of these serial experimental studies with E. coli pathogen model system for resistance development was to test our regulatory study hypothesis of whether HFIM can be applied to improve the drug development approval process. As we studied the resistance development against single and double or triple antibiotic combinations, our results indicated that triple antibiotic combinations significantly delayed the emergence of antibiotic resistance compared to single or double antibiotic combinations. We also would like to study the other treatment strategy of sequential treatment and its effect on resistant development. We provided more information on combination therapy and other factors affecting the antibiotic resistance development (See page 10).   

3. In all figures, I think the Y-axis should be labeled by CFU/mL if the log10 were added as the present manuscript, the number Y-axis should be 0, 1, 2, …, 12. It is appropriate to have two subgraphs per row.

Many thanks. This is now corrected to reflect CFU/mL in the Y axis (See figures in pages 4, 6, 7 and 8).

4. A, B and C, D, E, and F were all missed in the subplot of Figures. Further, why the geometric standard deviation of total population in some plots were much smaller than their subcomponents? What do these results mean?

Thank you for your comments. It looks like the figure labels A, B, C, D, E and F were missing during typesetting of the manuscript in the editorial manager. We hope that the revised figure is more accessible.

Individual hollow fiber (HF) experiments were conducted four times for simultaneous combination study and three times for sequential treatment studies. The individual experiment results were combined for simultaneous and sequential studies separately and then their geometric means were calculated. Each individual HF experiments were conducted with the culture made from a single colony and in a similar experimental condition. The variation in total viable population observed between experiments may be due to the nature of bacterial population at the time of study, combination, or sequential antibiotic treatment effects as well as the inherent experimental variation associated with the study. The variation in the total viable population observed in combination experiments may also be due to initial killing of bacteria up to 72 h and then different stages of resistance evolution at this time forward until 240 h.   

5. Format should be checked throughout the manuscript. Some places have extra Spaces or some sentences lack periods. Line 16, line 31 and line 33 in Page 8, line 82 in page 9, author names were not correctly listed (reference list 14, 18, 19).

We followed the advice of the reviewer and corrected throughout in the revised manuscript.

Reviewer 2 Report

The authors reported an evaluation of a sequential antibiotic treatment regimen of am-2 picillin, ciprofloxacin and fosfomycin against Escherichia coli 3 CFT073 in the hollow fiber infection model compared with 4 simultaneous combination treatments. There are the following comment as follows:

1. Introduction needs to be improved.

2. The rationale of the research must be included in the last section of the introduction in form of Fig.  

3.  Why did the authors only perform the study on E. coli instead of other bacterial stains?

4. Is there any correlation between these antibiotics' resistance with microorganism biofilm formation? 

5. Conclusion must be supported by the proposed aim hypothesis. 

Author Response

The authors would like to thank the reviewers for their valuable comments to improve the quality of the manuscript. The reviewer’s concerns were considered and incorporated changes accordingly in the revised manuscript. The changes in the revised manuscript were highlighted by track changes.

1. Introduction needs to be improved.

Many thanks for this comment. We now included more information on combination and sequential antibiotic treatments in the revised manuscripts. The information’s are related to advantage of using antibiotic combinations compared to monotherapy, some references where combinations showed synergistic effects against different bacterial species and mechanisms of resistance evolution. Also included information of recent studies on sequential antibiotic treatment (See page 2). 

2. The rationale of the research must be included in the last section of the introduction in form of Fig.

Many thanks for this suggestion. We now included a figure showing rationale of research in the last section of introduction in the revised manuscript (See page 3).

3. Why did the authors only perform the study on E. coli instead of other bacterial strains?

Many thanks for this question. We used E. coli in the hollow fiber infection model (HFIM) as an established pathogen model system. This pathogen model system will allow us to test our regulatory study hypothesis of whether HFIM will be useful model to improve the drug development approval process. If we determine that the HFIM proves to be a useful system for the drug development approval process, we will pursue studies with other bacterial strains.

4. Is there any correlation between these antibiotics’ resistance with microorganism biofilm formation?

This is indeed a very interesting topic. As our research focus is on utilizing HFIM to study the resistance development against antibiotics and whether this will be a useful model to improve the drug approval process. So, studying antibiotics and its resistance development of microorganism with biofilm formation is out of scope of our research.     

5. Conclusion must be supported by the proposed aim hypothesis.

Many thanks for your suggestions. The conclusion section is revised to reflect the study hypothesis (See page 12).

Round 2

Reviewer 1 Report

All suggestions were considered in this new edition. 

Reviewer 2 Report

The revised manuscript is suitable for further process in journal.